# Design, Synthesis, and In Vivo and In Silico Evaluation of Coumarin Derivatives with Potential Antidepressant Effects

**DOI:** 10.3390/molecules26185556

**Published:** 2021-09-13

**Authors:** Xuekun Wang, Hao Zhou, Xinyu Wang, Kang Lei, Shiben Wang

**Affiliations:** School of Pharmaceutical Sciences, Liaocheng University, Liaocheng 252059, China; lcuyxyzhouhao@163.com (H.Z.); wangxinyuWXY1128@163.com (X.W.); leikang@lcu.edu.cn (K.L.)

**Keywords:** coumarin, synthesis, antidepressant, FST, 5-HT, molecular docking studies

## Abstract

In this study, a series of coumarin derivatives were designed and synthesized, their structures were characterized using nuclear magnetic resonance (NMR) and high-resolution mass spectrometry (HRMS) testing methods. In the pharmacological experiment, two behavior-monitoring methods, the forced swim test (FST) and the tail suspension test (TST), were used to determine the antidepressant activity of coumarin derivatives. Compounds that showed potential activity were analyzed for their effects on 5-hydroxytryptamine (5-HT) levels in the brains of mice. Molecular docking experiments to simulate the possible interaction of these compounds with the 5-HT_1A_ receptor was also be predicted. The results of the pharmacological experiments showed that most coumarin derivatives exhibited significant antidepressant activity. Among these compounds, 7-(2-(4-(4-fluorobenzyl)piperazin-1-yl)-2-oxoethoxy)-2*H*-chromen-2-one (6i) showed the highest antidepressant activity. The results of the measurement of 5-HT levels in the brains of mice indicate that the antidepressant activity of coumarin derivatives may be mediated by elevated 5-HT levels. The results of molecular docking demonstrated that compound 6i had a significant interaction with amino acids around the active site of the 5-HT_1A_ receptor in the homology model. The physicochemical and pharmacokinetic properties of the target compounds were also predicted using Discovery Studio (DS) 2020 and Chemdraw 14.0.

## 1. Introduction

Many neurotransmitter systems are known to cause CNS diseases [1,2,3]. One of the common dysfunctions of the CNS is depression, whose occurrence is connected with disturbances in the functions of the serotonin and dopamine systems. Currently, drug therapy is the main method of depression treatment. These drugs mainly increase the concentration of serotonin, norepinephrine, or dopamine at the synapse by inhibiting membrane transporters or blocking the degradation of these neurotransmitters by monoamine oxidase. However, these drugs have certain adverse reactions and low efficacy [4,5]. Therefore, the development of novel antidepressants with high efficiency and low toxicity is an effective strategy for the treatment of depression.

It is well known that compounds containing piperazine structural fragments have important effects on biological activity; for example, they have been incorporated into the structures of compounds with anticancer, analgesic and antidiabetic activities [6,7,8]. Also, their antidepressant effects have been widely documented, for example, the marketed drugs Amooxapine, Sertraline, and Vilazodone (Figure 1) have piperazine fragments in their chemical structure. It has been reported that the introduction of piperazine fragments into appropriate compounds can also significantly improve the antidepressant activity of those compounds [9,10].

Over the years, coumarin has become an important core structure in drug design and has been widely used to synthesize compounds with anti-inflammatory, anti-Alzheimer, antioxidant, antimicrobial, anticancer, and antiepileptic activities [11,12,13,14,15,16]. However, its antidepressant effects have not been widely documented. Sashidhara et al. [17], designed and synthesized a series of amine-substituted 3-phenylcoumarin derivatives as potential antidepressant agents. Their studies demonstrate that the new coumarin derivatives may serve as a promising antidepressant lead and hence pave the way for further investigation around this chemical space (Figure 1A).

In our previous studies, we designed and synthesized a series of coumarin (Figure 1C) derivatives [18]. The results of the measurements of 5-HT levels in the brains of mice indicate that the antidepressant activity of coumarin derivatives may be mediated by elevated 5-HT levels. In this study, we combined structural fragments of piperazine and coumarin to improve their efficacy on depression, a series of 7-(2-(4-substitutedpiperazin-1-yl)-2-oxoethoxy)-2*H*-chromen-2-ones (**6a**–**u**) were designed and synthesized, and their antidepressant activities were tested.

The newly synthesized target compounds were characterized by the HRMS, ^1^H-, and ^13^C-NMR techniques. The antidepressant activities of the target compounds were evaluated using the FST and TST. The most active compound was used to evaluate the exploratory activity of the animals by the open-field test (OFT). Potentially active compounds were analyzed for their effects on 5-HT levels in the brains of mice. A homology model of the 5-HT_1A_ receptor was built using DS 2020 software and used to perform a molecular docking study. The physicochemical and pharmacokinetic properties of newly synthesized compounds were also predicted.

## 2. Results and Discussion

### 2.1. Chemistry

The synthetic route leading to target compounds **6a**–**u** is shown in Figure 1. Commercially available 1-boc piperazine (1) reacted at ice-bath temperature with chloroacetyl chloride in the presence of the base, to afford 2 in 70–90% yield. During the addition of chloroacetyl chloride to 1 in dichloromethane, the temperature of the reaction mixture should not exceed 5 °C, otherwise the yield of the piperazinecarboxylate 2 is negatively affected. Substitution of the chlorine atom in 2 by the hydroxyl group of commercially available 7-hydroxycoumarin (3) provided the ether derivative (4) in 75–90% yield. Deprotection of the boc group in 4 with excess trifluoroacetic acid in dichloromethane gave the monosubstituted piperazine derivative 5 in 70–90% yield. Crude derivative 5 was reacted with different alkyl (C_4_-C_9_) or substituted benzyl (Cl, F, CF_3_, CH_3_) compounds in acetonitrile containing potassium carbonate and potassium iodide, to afford products **6a**–**u**, after purification by column chromatography. The structures of all target compounds were confirmed by ^1^H-NMR, ^13^C-NMR, and HRMS technology.

### 2.2. Pharmacology

#### 2.2.1. FST

The forced swim test is the most commonly used validated rodent model to assess the antidepressant effect of compounds. In this study, we used the FST method to peform a preliminary evaluation the antidepressant activity of the synthesized target compound, and then screened out antidepressant compounds with potential. In this experiment, we used the 5-HT reuptake inhibitor fluoxetine, which has good clinical antidepressant activity and widely used as a reference drug for antidepressant screening [19]. Fluoxetine (positive control, 40 mg/kg) and target compounds **6a**–**u** (40 mg/kg) were administered intraperitoneally 0.5 h before the test sessions. The pharmacological data of the antidepressant activity of the synthesized target compounds obtained via the FST method are shown in Table 1.

As shown in Table 1, both the target compound and positive fluoxetine were administered at a dose of 40 mg/kg. It can be seen from the table that most of the compounds showed antidepressant activity, among which compounds 6b, 6d, 6j, 6k, 6n, 6q–r, and 6t showed significant differences compared with the control group (0.01 < *p* < 0.05). The three compounds 6c, 6i, and 6p showed significant differences compared with the control group (*p* < 0.01). Among all the compounds, compound 6i showed the best antidepressant activity, shortening the immobility time of mice by 84.2 s, which was slightly weaker than that of fluoxetine (immobility time was 84.13 s). The percentage decreases in the immobility duration (DID %) were 41.24 (6i) and 42.49 (fluoxetine), respectively. The percentage decrease in the immobility duration (DID %) was calculated using the formula: DID % = [(A-B)/A]∗100%. Where A is the duration of immobility (s) for the control group and B is the duration of immobility (s) for the test group.

To further investigate the effect of 6i and fluoxetine on antidepressant activity at different doses (10, 20, and 40 mg/kg, respectively), they were selected for the FST experiment. The results of the pharmacological experiments are presented in Figure 2. The antidepressant activity of compound 6i was different at the three doses and showed the best antidepressant activity at a dose of 40 mg/kg (DID = 31.54). Fluoxetine also showed good antidepressant activity at different doses, which was significantly different from that of the control group (*p* < 0.01, at a dose of 40 mg/kg, DID = 37.46). Based on these results, we speculated that the antidepressant activity of compound 6i may be dose-dependent, and as the dose increases, its antidepressant activity gradually increases.

#### 2.2.2. TST

TST is another classic method that can quickly evaluate the efficacy of antidepressants. In the FST model, compound 6i, with the best activity, was selected for the TST experiment to further investigate whether the compound still has antidepressant activity. The dosages of compound 6i and positive fluoxetine in the experiment were administered at 40 mg/kg, and the antidepressant activity results are shown in Figure 3. The pharmacological results showed that compound 6i also had a significant difference compared with the control group (0.01 < *p* < 0.05, DID value is 26.09), which is similar to that of the positive fluoxetine (0.01 < *p* < 0.05, DID value is 28.80).

#### 2.2.3. OFT

OFT is a classic animal experimental model that mainly observes the autonomous behavior, exploratory behavior, and tension of experimental animals in a new environment. The reduced immobility time in animal models of behavioral despair and depression may be caused by the excitement of the drug on the sympathetic nerves [20,21]. In this study, compound 6i was used in the OFT to evaluate the central excitability, and to determine whether compound 6i affects the spontaneous motor activity of mice. Pharmacological results are shown in Figure 4. Compared with the control group, compound 6i showed no significant difference (*p* > 0.05, motor activity: crossing, rearing, and grooming). Thus, these findings exclude any false-positive results attributed to central activity excitability.

#### 2.2.4. Determination of 5-HT Concentration

The abnormal 5-HT function of the central nervous system may be related to anorexia, tension, migraine, depression, schizophrenia, epilepsy, Parkinson, and other neuropsychiatric diseases. The monoamine hypothesis of depression is based on the fact that the lack of monoamine transmitter 5-HT in the brain is the biological basis of its pathogenesis. Most antidepressants exert pharmacological effects by increasing the levels of monoamine transmitters at synapses; for example, the selective 5-HT reuptake inhibitors fluoxetine and paroxetine produce antidepressant effects by increasing the content of 5-HT in the synapse and enhancing the neurotransmission function of 5-HT [22]. In this experiment, the enzyme-linked immunosorbent assay (ELISA) method was used to determine the effect of compound 6i on 5-HT content in mice brain tissue. The pharmacological experimental results are presented in Figure 5. Compared with the control group, compound 6i (40 mg/kg) significantly increased the 5-HT content in mice brain tissue, which was slightly weaker than that of fluoxetine (40 mg/kg).

#### 2.2.5. Structure–Activity Relationships

The structure–activity relationships (SAR_S_) of the target compounds were discussed based on the FST experimental results in Table 1. Among 6a–f, compounds 6a–c were introduced to a different alkyl group from C_4_H_9_ to C_6_H_13_. The length of the alkyl chain appeared to have an impact on antidepressant activity of the derivatives. From C_6_H_13_ to C_9_H_19_, as the alkyl chain length increased, antidepressant activity gradually increased (C_6_ > C_5_ > C_4_). From 6c to 6f, as the alkyl chain length increased, antidepressant activity decreased (C_6_ > C_7_> C_8_ > C_9_). Compound 6c (R = C_6_H_13_) was the most active compound. Obviously, in this study the activity curve of the alkyl chain substituted derivatives is bell-shaped with a maximum activity peak. Compound 6c, with the maximum activity peak, probably reflects the optimal partition coefficient associated with the easiest crossing of the biological membranes.

Among 6g–u, compounds containing electron-withdrawing groups (F, Cl, CF_3_) showed antidepressant activity in the order of *p*-F > *2,6*-F > *o*-F > *m*-F*, m*-Cl > *p*-Cl > *2,6*-Cl > *o*-Cl, *m*-CF_3_ > *o*-CF_3_ > *p*-CF_3_. By contrast, in the presence of the electron-donating group CH_3_, the activity order was *o*-CH_3_ > *m*-CH_3_ > *p*-CH_3_, while CH_3_ substitution in the *o*-position also led to the highest compound activity (Figure 6).

#### 2.2.6. Docking Study

This experiment used the most widely used molecular docking software, DS 2020, for molecular docking research. The 5-HT_1A_ receptor is a subtype of the serotonin receptor family. Studies have shown that 5-HT_1A_ is closely related to depression and has been widely used as a target in the field of antidepressant research. In this study, the 5-HT_1A_ receptor, which was constructed using the DS software and 6i/serotonin, were used for molecular docking research. The docking results are shown in Figure 7.

According to the literature [23], the amino acid residues around the active site of 5-HT_1A_ receptors, such as Gln97, Gly382, Ser199, Val117, Phe112, Phe361, Phe362, Thr121, Thr196, Thr200, Thr379, Ala93, Ala263, Ala365, Ala383, Asn386, Asp116, Cys120, Ile113, Ile124, and Ile167 are the key amino acid residues that interact with the ligand. As shown in Figure 7, compound 6i and serotonin can form an interactive force with some amino acids around the active site of the 5-HT_1A_ receptor. For example, compound 6i forms a classic hydrogen bond interaction with Thr121, Tyr390, Ala93, and Leu381. The benzene ring of compound 6i forms a π-π interaction with Phe361 and Phe362. Serotonin forms a classic hydrogen bond interaction with Asn386. The benzene ring of serotonin forms a π-π interaction with Phe361 and Phe362. Based on the interaction between compound 6i and the amino acid residues around the active pocket of the 5-HT_1A_ receptor, such as hydrophobic and hydrogen-bonding interactions, it is speculated that compound 6i exerts antidepressant activity and is related to the 5-HT_1A_ receptor.

#### 2.2.7. Prediction of Pharmacokinetic and Physicochemical Properties

Regarding the introduction of a drug to society, especially the marketing prospect, is not enough to have strong curative effects and low side effects; it must also have good pharmacokinetic properties. In this study, the ADME/T module in the DS 2020 software package was used to predict the pharmacokinetic properties of target compounds **6a**–**u**, such as blood-brain barrier (BBB) permeability and absorption (ABS), to avoid the compounds’ difficulty in penetrating the BBB, low oral bioavailability, and low solubility. The prediction results showed that all target compounds exhibited favorable BBB permeability and good ABS (Figure 8).

Lipinski’s “Rule of Five,” including such as molecular weight (MW ≤ 500), number of hydrogen donors (nHBD ≤ 5), number of rotatable bonds (RotB ≤ 10), LogP ≤ 5, and the number of acceptors (nHBA ≤ 10), is usually used in drug design and screening. In fact, a compound that meets Lipinski’s “Rule of Five” is more likely to be a suitable drug, even in the case of antidepressant drugs. DS 2020 and ChemBioDraw Ultra 14.0 were used to construct the target compound’s structural formula and predict physicochemical properties, such as MW, RotB, CLogP, nHBD, nHBA. As shown in Table 2, almost all target compounds met Lipinski’s “Rule of Five.” Simultaneously, the polar surface area (TPSA) of the target compounds was also predicted, and the TPSA of all target compounds was shown to be less than 140 (TPSA > 140 is considered to indicate low oral bioavailability) [24].

## 3. Materials and Methods

### 3.1. Chemistry

An X-4 melting point apparatus with a binocular microscope was used to determine the melting point (m.p.) of the target compound **6a**–**u**. HRMS were obtained using mass spectrometry (Bruker Daltonik, Germany). Using tetramethylsilane as the internal standard, the ^1^H-NMR and ^13^C-NMR data of the target compound were measured using a Bruker AV-500 nuclear magnetic resonance instrument (compounds **6a**–**u** were dissolved in CDCl_3_ or DMSO*_d6_*). The positive fluoxetine used in the pharmacological experiment was of a high-purity standard which can be directly used in pharmacological experiments. All chemical reagents used in the chemical experiment were of analytical reagent grade or chemically pure, which were directly used in the experiment without further purification. The NMR and HRMS spectra of compounds 4, 5, and **6a**–**u** are presented in Appendix A.

#### 3.1.1. General Procedure for the Synthesis of tert-butyl 4-(2-chloroacetyl)piperazine-1-carboxylate 2

1-Boc piperazine (1.86 g, 10.0 mmol) and potassium carbonate (1.66 g, 12.0 mmol) were added to dichloromethane (30 mL), and stirred at room temperature for 0.5 h. The reaction mixture was cooled to 0 °C using an ice bath and then chloroacetyl chlorider (1.23 g, 11.0 mmol) was added dropwise in order to maintain the temperature at 0–5 °C. After the reaction was complete (TLC examination), the solvent was evaporated under reduced pressure, the residue was dissolved in water (30 mL), and filtered off (washed with water 3 times × 10 mL) under reduced pressure to give product 2 as a solid in 70–99% yield. The product was used in the next step without further purification.

#### 3.1.2. General Procedure for the Synthesis of tert-nbutyl 4-{2-[(2-oxo-2H-chromen-7-yl)oxy]acetyl}piperazine-1-carboxylate 4

7-Hydroxy-2*H*-chromen-2-one 3 (1.62 g, 10.0 mmol), piperazine-1-carboxylate 2 (2.9 g, 11.0 mmol), potassium carbonate (1.52 g, 11.0 mmol) and a catalytic amount of potassium iodide were added to acetone (30 mL). The resulting mixture was stirred under reflux at 80 °C for 25–30 h. After the reaction was complete (TLC examination), the solvent was evaporated under reduced pressure. The residue was dissolved in 30 mL water, filtered with suction, washed three times with water, and dried to obtain product 4. The product was used in the next step without further purification.

White solid, yield 75–90%. m.p. 169~171 °C; ^1^H NMR (500 MHz, CDCl_3_) δ: 6.27–7.65 (m, Coumarin-H, 5H), 4.79 (s, -OCH_2_, 2H), 3.42–3.61 (m, Piperazine-H, 8H), 1.47 (s, -(CH_3_)_3_, 9H); ^13^C NMR (125 MHz, CDCl_3_) δ: 165.55, 160.88, 160.79, 155.69, 154.44, 143.19, 129.09, 113.84, 113.43, 112.56, 102.04, 80.55, 67.47, 52.87, 45.16, 42.00, 28.40, 28.40.

#### 3.1.3. General Procedure for the Synthesis of 7-[2-oxo-2-(piperazin-1-yl)ethoxy]-2H-chromen-2-one 5

Tert-butyl 4-{2-[(2-oxo-2*H*-chromen-7-yl)oxy]acetyl}piperazine-1-carboxylate 4 (3.89 g, 10.0 mmol) and trifluoroacetic acid (1 mL, 13.4 mmol) were added to dichloromethane (30 mL) and the reaction was stirred at room temperature for 1 h. TLC examination indicated the reaction was complete. The excess dichloromethane and trifluoroacetic acid were evaporated under vacuo to obtain product 5. The product was used in the next step without further purification.

Brown oil, yield 70–90%; ^1^H NMR (500 MHz, DMSO*_d6_*) δ: 6.30–8.01 (m, Coumarin-H, 5H), 5.04k (s, -OCH_2_, 2H), 3.11–3.21, 3.64–3.66 (m, Piperazine-H, 8H), 1.99 (s, NH, 1H); ^13^C NMR (125 MHz, DMSO*_d6_*) δ: 164.07, 160.03, 159.93, 154.66, 142.22, 127.98, 112.66, 112.27, 111.63, 101.05, 66.23, 51.17, 44.17, 41.14.

#### 3.1.4. General Procedure for the Synthesis of Target Compounds **6a**–**u**

7-(2-Oxo-2-(piperazin-1-yl)ethoxy)-2*H*-chromen-2-one **5** (2.9 g, 10.0 mmol), potassium carbonate (1.52 g, 11.0 mmol) and a catalytic amount of potassium iodide were added to acetonitrile (30 mL) and the reaction mixture was refluxed for 0.5 h. The appropriate substituted alkyl (10.0 mmol, C_4_-C_9_) or benzyl (10.0 mmol, Cl, F, CF_3_, CH_3_) bromide was then added and refluxing was continued for 24 h. After the reaction was complete as indicated by TLC examination, the solvent was evaporated under reduced pressure to give a residue, to which water (30 mL) was added. The resulting suspension was extracted with dichloromethane (3 × 10 mL) and the combined solvents dried by MgSO_4_. The drying agent was filtered off and the solvent was removed under reduced pressure to give the appropriate crude products **6a**–**u** which were purified using column chromatography (methanol:dichloromethane, 1:50) to give the appropriate target compounds **6a**–**u**. The yields, melting points, and spectral data (^1^H-NMR, ^13^C-NMR and HRMS) of compounds **6a**–**u** are given below.

7-(2-(4-Butylpiperazin-1-yl)-2-oxoethoxy)-2*H*-chromen-2-one (6a): White solid, yield 72.1%. m.p. 139~140 °C; ^1^H NMR (500 MHz, CDCl_3_) δ: 6.19–7.58 (m, Coumarin-H, 5H), 4.70 (s, -OCH_2_, 2H), 2.34–2.40, 3.49–3.58 (m, Piperazine-H, 8H), 2.26–2.29 (t, -CH_2_, J = 7.5 Hz, 2H), 1.22–1.43 (m, -(CH_2_)_2_-, 4H), 0.83–0.85 (t, -CH_3_, J = 7.5 Hz, 3H); ^13^C NMR (125 MHz, CDCl_3_) δ: 164.07, 160.03, 159.93, 154.66, 142.22, 127.98, 112.66, 112.27, 111.63, 101.05, 66.23, 57.19, 52.31, 51.71, 44.17, 41.14, 27.88, 19.59, 12.98; HRMS calcd for C_19_H_25_N_2_O_4_^+^ [M + H]^+^ 345.1809, found 345.1806.

7-(2-Oxo-2-(4-pentylpiperazin-1-yl)ethoxy)-2*H*-chromen-2-one (6b): White solid, yield 64.5%. m.p. 130~131 °C; ^1^H NMR (500 MHz, CDCl_3_) δ: 6.19–7.58 (m, Coumarin-H, 5H), 4.70 (s, -OCH_2_, 2H), 2.34–2.40, 3.47–3.58 (m, Piperazine-H, 8H), 2.25–2.28 (t, -CH_2_, J = 7.5 Hz, 2H), 1.21–1.41 (m, -(CH_2_)_3_-, 6H), 0.81–0.84 (t, -CH_3_, J = 7.5 Hz, 3H); ^13^C NMR (125 MHz, CDCl_3_) δ: 164.07, 160.02, 159.93, 154.66, 142.22, 127.97, 112.66, 112.27, 111.63, 101.05, 66.23, 57.49, 52.31, 51.71, 44.16, 41.14, 28.61, 25.44, 21.55, 13.01; HRMS calcd for C_20_H_27_N_2_O_4_^+^ [M+H]^+^ 359.1965, found 359.1961.

7-(2-(4-Hexylpiperazin-1-yl)-2-oxoethoxy)-2*H*-chromen-2-one (6c): White solid, yield 72.1%. m.p. 118~119 °C; ^1^H NMR (500 MHz, CDCl_3_) δ: 6.26–7.65 (m, Coumarin-H, 5H), 4.77 (s, -OCH_2_, 2H), 2.41–2.47, 3.54–3.66 (m, Piperazine-H, 8H), 2.33–2.36 (t, -CH_2_, J = 7.5 Hz, 2H), 1.29–1.49 (m, -(CH_2_)_4_-, 8H), 0.87–0.90 (t, -CH_3_, J = 7.5 Hz, 3H); ^13^C NMR (125 MHz, CDCl_3_) δ: 165.09, 161.04, 160.94, 155.68, 143.23, 128.99, 113.68, 113.29, 112.65, 102.07, 67.25, 58.54, 53.32, 52.73, 45.18, 42.15, 31.73, 27.12, 26.73, 22.59, 14.04; HRMS calcd for C_21_H_29_N_2_O_4_^+^ [M+H]^+^ 373.2122, found 373.2120.

7-(2-(4-Heptylpiperazin-1-yl)-2-oxoethoxy)-2*H*-chromen-2-one (6d): White solid, yield 55.9%. m.p. 121~122 °C; ^1^H NMR (500 MHz, CDCl_3_) δ: 6.26–7.65 (m, Coumarin-H, 5H), 4.77 (s, -OCH_2_, 2H), 2.42–2.47, 3.54–3.66 (m, Piperazine-H, 8H), 2.33–2.36 (t, -CH_2_, J = 7.5 Hz, 2H), 1.29–1.49 (m, -(CH_2_)_5_-, 10H), 0.87–0.90 (t, -CH_3_, J = 7.5 Hz, 3H); ^13^C NMR (125 MHz, CDCl_3_) δ: 165.09, 161.04, 160.94, 155.68, 143.23, 128.99, 113.69, 113.29, 112.65, 102.07, 67.25, 58.53, 53.32, 52.72, 45.17, 42.14, 31.78, 29.18, 27.40, 26.76, 22.61, 14.08; HRMS calcd for C_22_H_31_N_2_O_4_^+^ [M+H]^+^ 387.2278, found 387.2277.

7-(2-(4-Octylpiperazin-1-yl)-2-oxoethoxy)-2*H*-chromen-2-one (6e): White solid, yield 66.4%. m.p. 124~125 °C; ^1^H NMR (500 MHz, CDCl_3_) δ: 6.26–7.65 (m, Coumarin-H, 5H), 4.77 (s, -OCH_2_, 2H), 2.42–2.46, 3.55–3.65 (m, Piperazine-H, 8H), 2.32–2.35 (t, -CH_2_, J = 7.5 Hz, 2H), 1.28–1.47 (m, -(CH_2_)_6_-, 12H), 0.87–0.90 (t, -CH_3_, J = 7.5 Hz, 3H); ^13^C NMR (125 MHz, CDCl_3_) δ: 165.09, 161.04, 160.94, 155.69, 143.22, 128.99, 113.71, 113.29, 112.64, 102.08, 67.27, 58.55, 53.33, 52.73, 45.18, 42.16, 31.82, 29.49, 29.24, 27.45, 26.77, 22.65, 14.10; HRMS calcd for C_23_H_33_N_2_O_4_^+^ [M+H]^+^ 401.2435, found 401.2432.

7-(2-(4-Nonylpiperazin-1-yl)-2-oxoethoxy)-2*H*-chromen-2-one (6f): White solid, yield 72.3%. m.p. 119~120 °C; ^1^H NMR (500 MHz, CDCl_3_) δ: 6.19–7.58 (m, Coumarin-H, 5H), 4.70 (s, -OCH_2_, 2H), 2.34–2.40, 3.47–3.58 (m, Piperazine-H, 8H), 2.25–2.28 (t, -CH_2_, J = 7.5 Hz, 2H), 1.19–1.42 (m, -(CH_2_)_7_-, 14H), 0.79–0.82 (t, -CH_3_, J = 7.5 Hz, 3H); ^13^C NMR (125 MHz, CDCl_3_) δ: 164.08, 160.02, 159.93, 154.66, 142.22, 127.98, 112.67, 112.27, 111.64, 101.05, 66.23, 57.52, 52.30, 51.71, 44.15, 41.13, 30.85, 28.51, 28.25, 26.43, 25.74, 21.65, 13.09; HRMS calcd for C_24_H_35_N_2_O_4_^+^ [M+H]^+^ 415.2591, found 415.2587.

7-(2-(4-(2-Fluorobenzyl)piperazin-1-yl)-2-oxoethoxy)-2*H*-chromen-2-one (6g): White solid, yield 68.1%. m.p. 136~137 °C; ^1^H NMR (500 MHz, CDCl_3_) δ: 6.19–7.57 (m, Coumarin-H and Ar-H, 9H), 4.69 (s, -OCH_2_, 2H), 3.55 (s, -CH_2_-, 2H), 2.42–2.44, 3.48–3.58 (m, Piperazine-H, 8H); ^13^C NMR (125 MHz, CDCl_3_) δ: 164.15, 161.37, 159.97, 159.94, 159.41, 154.65, 142.22, 130.53, 128.16, 127.98, 123.00, 122.97, 114.48, 114.31, 112.68, 112.29, 111.59, 101.06, 66.25, 54.08, 51.71, 51.29, 49.80, 44.11, 41.08; HRMS calcd for C_22_H_22_FN_2_O_4_^+^ [M+H]^+^ 397.1558, found 397.1556.

7-(2-(4-(3-Fluorobenzyl)piperazin-1-yl)-2-oxoethoxy)-2*H*-chromen-2-one (6h): White solid, yield 77.7%. m.p. 137~138 °C; ^1^H NMR (500 MHz, CDCl_3_) δ: 6.19–7.58 (m, Coumarin-H and Ar-H, 9H), 4.70 (s, -OCH_2_, 2H), 3.45 (s, -CH_2_-, 2H), 2.37–2.41, 3.47–3.58 (m, Piperazine-H, 8H); ^13^C NMR (125 MHz, CDCl_3_) δ: 164.24, 162.95, 160.99, 160.03, 159.97, 154.64, 142.29, 128.87, 128.80, 128.02, 123.51, 123.49, 114.75, 114.58, 113.38, 113.21, 112.67, 112.31, 111.67, 101.01, 66.23, 61.12, 51.96, 51.56, 49.77, 44.12, 41.11; HRMS calcd for C_22_H_22_FN_2_O_4_^+^ [M+H]^+^ 397.1558, found 397.1557.

7-(2-(4-(4-Fluorobenzyl)piperazin-1-yl)-2-oxoethoxy)-2*H*-chromen-2-one (6i): White solid, yield 74.2%. m.p. 171~172 °C; ^1^H NMR (500 MHz, CDCl_3_) δ: 6.19–7.57 (m, Coumarin-H and Ar-H, 9H), 4.69 (s, -OCH_2_, 2H), 3.41 (s, -CH_2_-, 2H), 2.36–2.37, 3.47–3.57 (m, Piperazine-H, 8H); ^13^C NMR (125 MHz, CDCl_3_) δ: 164.17, 162.11, 160.16, 159.98, 159.95, 154.65, 142.23, 132.13, 129.58, 129.52, 128.00, 114.28, 114.11, 112.68, 112.29, 111.63, 101.01, 66.24, 60.91, 51.90, 51.49, 49.80, 44.13, 41.11; HRMS calcd for C_22_H_22_FN_2_O_4_^+^ [M+H]^+^ 397.1558, found 397.1553.

7-(2-(4-(2-Chlorobenzyl)piperazin-1-yl)-2-oxoethoxy)-2*H*-chromen-2-one (6j): White solid, yield 69.5%. m.p. 120~121 °C; ^1^H NMR (500 MHz, CDCl_3_) δ: 6.18–7.57 (m, Coumarin-H and Ar-H, 9H), 4.70 (s, -OCH_2_, 2H), 3.57 (s, -CH_2_-, 2H), 2.43–2.48, 3.47–3.57 (m, Piperazine-H, 8H); ^13^C NMR (125 MHz, CDCl_3_) δ: 164.17, 159.99, 159.94, 154.65, 142.24, 134.07, 133.44, 129.75, 128.60, 127.99, 127.50, 125.68, 112.66, 112.28, 111.62, 101.03, 66.25, 58.04, 51.98, 51.57, 44.19, 41.16; HRMS calcd for C_22_H_22_ClN_2_O_4_^+^ [M+H]^+^ 413.1263, found 413.1262.

7-(2-(4-(3-Chlorobenzyl)piperazin-1-yl)-2-oxoethoxy)-2*H*-chromen-2-one (6k): White solid, yield 62.2%. m.p. 132~133 °C; ^1^H NMR (500 MHz, CDCl_3_) δ: 6.19–7.57 (m, Coumarin-H and Ar-H, 9H), 4.69 (s, -OCH_2_, 2H), 3.43 (s, -CH_2_-, 2H), 2.38, 3.48–3.58 (m, Piperazine-H, 8H); ^13^C NMR (125 MHz, CDCl_3_) δ: 164.17, 159.96, 159.93, 154.65, 142.22, 133.32, 128.64, 128.00, 127.94, 126.55, 126.08, 112.69, 112.29, 111.62, 101.02, 66.26, 61.08, 51.96, 51.58, 49.81, 44.13, 41.10; HRMS calcd for C_22_H_22_ClN_2_O_4_^+^ [M+H]^+^ 413.1263, found 413.1260.

7-(2-(4-(4-Chlorobenzyl)piperazin-1-yl)-2-oxoethoxy)-2*H*-chromen-2-one (6l): White solid, yield 64.8%. m.p. 143~144 °C; ^1^H NMR (500 MHz, CDCl_3_) δ: 6.26–7.65 (m, Coumarin-H and Ar-H, 9H), 4.77 (s, -OCH_2_, 2H), 3.48 (s, -CH_2_-, 2H), 2.44, 3.54–3.64 (m, Piperazine-H, 8H); ^13^C NMR (125 MHz, CDCl_3_) δ: 165.19, 160.98, 155.67, 143.25, 133.12, 130.33, 129.02, 128.55, 113.70, 113.31, 112.65, 102.03,67.26, 61.96, 52.95, 52.55, 50.83, 45.15, 42.12; HRMS calcd for C_22_H_22_ClN_2_O_4_^+^ [M+H]^+^ 413.1263, found 413.1263.

7-(2-Oxo-2-(4-(2-(trifluoromethyl)benzyl)piperazin-1-yl)ethoxy)-2*H*-chromen-2-one (6m): White solid, yield 54.3%. m.p. 134~135 °C; ^1^H NMR (500 MHz, CDCl_3_) δ: 6.19–7.69 (m, Coumarin-H and Ar-H, 9H), 4.70 (s, -OCH_2_, 2H), 3.61 (s, -CH_2_-, 2H), 2.41–2.43, 3.48–3.58 (m, Piperazine-H, 8H); ^13^C NMR (125 MHz, CDCl_3_) δ: 164.19, 159.98, 159.92, 154.68, 142.21, 135.93, 130.81, 129.39, 127.99, 127.91, 127.67, 126.14, 124.97, 124.92, 124.46, 122.28, 112.70, 112.30, 111.61, 101.04, 66.30, 57.11, 52.42, 52.09, 51.66, 44.23, 41.20; HRMS calcd for C_23_H_22_F_3_N_2_O_4_^+^ [M+H]^+^ 447.1526, found 447.1522.

7-(2-Oxo-2-(4-(3-(trifluoromethyl)benzyl)piperazin-1-yl)ethoxy)-2*H*-chromen-2-one (6n): White solid, yield 65.4%. m.p. 117~118 °C; ^1^H NMR (500 MHz, CDCl_3_) δ: 6.26–7.65 (m, Coumarin-H and Ar-H, 9H), 4.77 (s, -OCH_2_, 2H), 3.58 (s, -CH_2_-, 2H), 2.47, 3.56–3.66 (m, Piperazine-H, 8H); ^13^C NMR (125 MHz, CDCl_3_) δ: 165.20, 160.98, 160.93, 155.68, 143.23, 138.67, 132.27, 130.94, 130.69, 129.02, 128.88, 125.55, 125.22, 124.29, 123.06, 113.73, 113.32, 112.63, 102.04,67.30, 62.16, 53.44, 53.01, 52.63, 45.13, 42.09; HRMS calcd for C_23_H_22_F_3_N_2_O_4_^+^ [M+H]^+^ 447.1526, found 447.1525.

7-(2-Oxo-2-(4-(4-(trifluoromethyl)benzyl)piperazin-1-yl)ethoxy)-2*H*-chromen-2-one (6o): White solid, yield 62.5%. m.p. 145~146 °C; ^1^H NMR (500 MHz, CDCl_3_) δ: 6.18–7.57 (m, Coumarin-H and Ar-H, 9H), 4.70 (s, -OCH_2_, 2H), 3.50 (s, -CH_2_-, 2H), 2.38–2.40, 3.47–3.58 (m, Piperazine-H, 8H); ^13^C NMR (125 MHz, CDCl_3_) δ: 165.15, 161.03, 160.98, 155.67, 143.26, 137.02, 134.31, 129.09, 129.06, 129.00, 113.67, 113.29, 112.66, 102.06,67.24, 62.52, 52.98, 52.54, 50.81, 45.19, 42.18, 21.11; HRMS calcd for C_23_H_22_F_3_N_2_O_4_^+^ [M+H]^+^ 447.1526, found 447.1524.

7-(2-(4-(2-Methylbenzyl)piperazin-1-yl)-2-oxoethoxy)-2*H*-chromen-2-one (6p): White solid, yield 67.1%. m.p. 113~114 °C; ^1^H NMR (500 MHz, CDCl_3_) δ: 6.18–7.57 (m, Coumarin-H and Ar-H, 9H), 4.69 (s, -OCH_2_, 2H), 3.42 (s, -CH_2_-, 2H), 2.39, 3.44–3.55 (m, Piperazine-H, 8H), 2.29 (s,-CH_3_, 3H); ^13^C NMR (125 MHz, CDCl_3_) δ: 164.12, 160.02, 159.92, 154.67, 142.21, 136.58, 129.43, 128.91, 127.98, 126.42, 124.60, 112.67, 112.28, 111.63, 101.05, 66.26, 59.65, 52.03, 51.65, 44.22, 41.21, 18.21; HRMS calcd for C_23_H_25_N_2_O_4_^+^ [M+H]^+^ 393.1809, found 393.1806.

7-(2-(4-(3-Methylbenzyl)piperazin-1-yl)-2-oxoethoxy)-2*H*-chromen-2-one (6q): White solid, yield 70.5%. m.p. 93~94 °C; ^1^H NMR (500 MHz, CDCl_3_) δ: 6.18–7.57 (m, Coumarin-H and Ar-H, 9H), 4.69 (s, -OCH_2_, 2H), 3.41 (s, -CH_2_-, 2H), 2.39–2.39, 3.45–3.58 (m, Piperazine-H, 8H), 2.27 (s,-CH_3_, 3H); ^13^C NMR (125 MHz, CDCl_3_) δ: 164.12, 160.02, 159.91, 154.67, 142.20, 136.99, 136.32, 128.81, 127.97, 127.22, 127.08, 125.16, 112.68, 112.27, 111.62, 101.06, 66.26, 61.80, 52.03, 51.62, 44.18, 41.16, 20.37; HRMS calcd for C_23_H_25_N_2_O_4_^+^ [M+H]^+^ 393.1809, found 393.1805.

7-(2-(4-(4-Methylbenzyl)piperazin-1-yl)-2-oxoethoxy)-2*H*-chromen-2-one (6r): White solid, yield 54.2%. m.p. 128~129 °C; ^1^H NMR (500 MHz, CDCl_3_) δ: 6.26–7.64 (m, Coumarin-H and Ar-H, 9H), 4.76 (s, -OCH_2_, 2H), 3.48 (s, -CH_2_-, 2H), 2.42–2.46, 3.51–3.64 (m, Piperazine-H, 8H), 2.34 (s,-CH_3_, 3H); ^13^C NMR (125 MHz, CDCl_3_) δ: 165.15, 161.03, 160.98, 155.67, 143.26, 137.02, 134.31, 129.09, 129.06, 129.00, 113.67, 113.29, 112.66, 102.06,67.24, 62.52, 52.98, 52.54, 50.81, 45.19, 42.18, 21.11.; HRMS calcd for C_23_H_25_N_2_O_4_^+^ [M+H]^+^ 393.1809, found 393.1805.

7-(2-(4-Benzylpiperazin-1-yl)-2-oxoethoxy)-2*H*-chromen-2-one (6s): White solid, yield 58.8%. m.p. 158~159 °C; ^1^H NMR (500 MHz, CDCl_3_) δ: 6.26–7.64 (m, Coumarin-H and Ar-H, 9H), 4.76 (s, -OCH_2_, 2H), 3.53 (s, -CH_2_-, 2H), 2.44–2.48, 3.55–3.65 (m, Piperazine-H, 8H); ^13^C NMR (125 MHz, CDCl_3_) δ: 165.15, 161.02, 160.96, 155.68, 143.25, 137.42, 129.10, 129.00, 128.39, 127.38, 113.69, 113.29, 112.64, 102.06, 67.26, 62.78, 53.01, 52.60, 45.19, 42.17; HRMS calcd for C_22_H_23_N_2_H_4_^+^ [M+H]^+^ 379.1652, found 379.1650.

7-(2-(4-(2,6-Difluorobenzyl)piperazin-1-yl)-2-oxoethoxy)-2*H*-chromen-2-one (6t): White solid, yield 74.5%. m.p. 129~130 °C; ^1^H NMR (500 MHz, CDCl_3_) δ: 6.18–7.56 (m, Coumarin-H and Ar-H, 9H), 4.67 (s, -OCH_2_, 2H), 3.65 (s, -CH_2_-, 2H), 2.42–2.46, 3.45–3.57 (m, Piperazine-H, 8H); ^13^C NMR (125 MHz, CDCl_3_) δ: 164.08, 161.97, 161.91, 160.00, 159.96, 159.92, 154.63, 142.22, 128.68, 128.60, 128.52, 127.96, 112.65, 112.25, 111.55, 111.07, 110.30, 110.26, 110.14, 110.10, 101.06,66.20, 51.14, 50.68, 47.47, 44.11, 41.08; HRMS calcd for C_22_H_21_F_2_N_2_O_4_^+^ [M+H]^+^ 415.1464, found 415.1459.

7-(2-(4-(2,6-Dichlorobenzyl)piperazin-1-yl)-2-oxoethoxy)-2*H*-chromen-2-one (6u): White solid, yield 81.3%. m.p. 132~133 °C; ^1^H NMR (500 MHz, CDCl_3_) δ: 6.26–7.64 (m, Coumarin-H and Ar-H, 9H), 4.77 (s, -OCH_2_, 2H), 3.78 (s, -CH_2_-, 2H), 2.60, 3.51–3.61 (m, Piperazine-H, 8H); ^13^C NMR (125 MHz, CDCl_3_) δ: 165.11, 161.02, 160.95, 155.69, 143.23, 136.99, 129.17, 129.00, 128.47, 113.70, 113.30, 112.62, 102.10, 67.29, 56.18, 53.43, 52.85, 52.48, 45.23, 42.21; HRMS calcd for C_22_H_21_Cl_2_N_2_O_4_^+^ [M+H]^+^ 447.0873, found 447.0869.

### 3.2. Pharmacology

#### 3.2.1. Evaluation of In Vivo Antidepressant Activity

Kunming mice (body weight 18–22 g, mice could eat and drink freely before the experiment) were used to determine in vivo antidepressant activity for target compounds **6a**–**u**, and the marketed drug fluoxetine was used as the positive group. Before the test, both fluoxetine and the target compound were dissolved in dimethyl sulfoxide or 0.5% methylcellulose. Two traditional experimental models, FST [25,26] and TST [27,28], were used to initially screen the antidepressant activity of the target compound, and the target compound was administered via the intraperitoneal/oral injection methods. Then the compound with potential activity was selected for the OFT [29] to further test the effect of the compound on spontaneous activity in mice. The 5-HT concentration was measured using the enzyme-linked immunosorbent assay (ELISA) method to determine whether compound 6i showed an effect in the mice brain.

#### 3.2.2. In Silico Studies

The structures of the ligands were etched using ChemBioDraw Ultra 14.0. The CDOCKER molecular docking module in DS 2020 was used for molecular docking research on potential target compounds. The binding site sphere of the protein was defined (x = 4.093, y = 17.549, z = 21.125 and radius = 10), and selected the compound with a high score in the docking result for analysis. The FASTA sequence of the 5-HT_1A_ receptor was retrieved from RCSB-PDB (https://www.rcsb.org/). Using the homology-modeled 5-HT_1A_ crystal structure as a template (structure crystals with high homology PDB: 4IAR, 4IAQ, 5V54, and 6G79), the analysis module of the DS was used to analyze the results. DS 2020 and ChemBioDraw Ultra 14.0 were used to predict the physicochemical and pharmacokinetic properties of the target compound.

#### 3.2.3. Statistical Analysis

The data are expressed as the mean ± standard deviation. One-way analysis of variance was performed using GraphPad Prism 5.0 statistical software. 0.01 < *p* < 0.05 indicates significant difference. *p* < 0.01 indicates that the difference is very significant.

## 4. Conclusions

A library of 21 new coumarin derivatives **6a**–**u** has been synthesized and assessed for in vivo antidepressant activity. The chemical structure of these compounds was determined by ^1^H-NMR and ^13^C-NMR spectroscopy and HRMS spectrometry. The results of pharmacological experiments showed that most of the target compounds exhibited antidepressant activity in the FST antidepressant model. Among them, compound 6i showed the best antidepressant activity in FST and TST models, similar to positive fluoxetine (in the FST model, compound 6i and fluoxetine shortened the immobility time of mice by 84.2 s and 82.4 s, respectively; in the TST model: compound 6i and fluoxetine shortened the immobility time of mice by 90.3 s and 87.0 s, respectively). OFT results showed that compound 6i had no significant effect on spontaneous activity in mice. The results of the determination of 5-HT content showed that compound 6i can significantly increase its level in mice brains. The results of molecular docking showed that compound 6i interacts more with the amino acid residues around the active site of the 5-HT_1A_ receptor in the homology model. Therefore, the mode of action of compound 6i, supporting its antidepressant activity, may be closely related to the 5-HT_1A_ receptor. Prediction of the physicochemical and pharmacokinetic properties of the target compounds showed that all compounds had good physicochemical properties, as well as good BBB permeability and bioavailability.

## Data Availability

The data presented in this study are available on request from the corresponding author.

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
