# Peer review of "Design, Synthesis, and In Vivo and In Silico Evaluation of Coumarin Derivatives with Potential Antidepressant Effects"

_molecules, 2021, doi:10.3390/molecules26185556_

Round 1

Reviewer 1 Report

The manuscript entitled “Design, synthesis, and in vivo and in silico evaluation of coumarin derivatives as new antidepressant agents” deals with the design and the synthesis of a series of coumarin derivatives, their structural characterization and potential use as antidepressants. Molecular docking experiments were used to simulate the interaction of these compounds with the 5-HT1A receptor. This topic is very important and actual. The manuscript is very well written. The research is organized in a highly scientific manner, and the results are promising since one of the synthesized compounds has great potential in depression treatment. Since it is a known side-effect of many antidepressants cognitive impairment, I would recommend the authors discuss the influence of the synthesized compounds on the AChE and BuChe activity. It would be great to see the experimental results as well. It would add great value to this research, in my opinion. 

Author Response

Dear sir or madam,

Thank you very much for your valuable comments on my article, which is very helpful for my article improvement. I have made changes in accordance with your comments.

Reviewer 2 Report

It is clear that a series of piperazine-modified coumarins were synthesized and characterized, and several in vivo experiments were done, but to fully state that they are antidepressants, even if they affect 5-HT brain levels, appear to a non-expert excessive to declare it "antidepressant" in the title.

The authors base their interesting and comprehensive research on the correlation of chemical structure and specific in vivo tests called the Forced swimming test (FST), tail suspension test (TST), and an open-field test.  Also, provide more detail on how the FST test is done. The text in section 2.2.1 is scarce and impossible to understand to a more chemically oriented audience. In section 2.2.3 please provide a photo of the setup used to evaluate the conduct of the mice. It is hard to understand how the behavior could be measured quantitatively and obtain such numbers as shown in figure 4.

Please explain in detail how such in vivo tests are used in pharmacology in the early stages of antidepressant drug development.

Please provide evidence that the authors have complied with the permits and authorization of the bioethics committee for use of live animal experimentation.

Since "Molecules" is not a specialized journal in pharmacology, explain in detail how those FST, TST, and open field tests are done, and how such tests can provide enough evidence or a clinical correlate to declare that a compound is a "bona fide" antidepressant. 

With respect to the synthesis and chemical characterization of the coumarin derivatives I have no problem, the work appears properly done. 

Please provide more detail on the in silico studies sec 3.2.2 More details on the docking and building of the 5-HT receptor. Also, how the docking results were scored?

Conclusions need to be expanded and also contrasted with literature, in which molecules like fluoxetine or other demonstrated antidepressant drugs score on in vivo assays. Also, what are the following steps in the development of antidepressant drugs? How one goes towards testing in humans? Toxicity tests? Although it may be speculative, that will provide a perspèctive to the non-expert audience with a more chemical background.

Author Response

(The authors gave the same response as above.)

Reviewer 3 Report

The manuscript describes interesting work on the synthesis, characterization and biological activity of 21 novel 7-(2-(4-alkylpiperazin-1-yl)-2-oxoethoxy)-2H-chromen-2-ones. 

However, throughout the manuscript there are quite a number of English grammar and editorial mistakes. The experimental procedures are poorly written and compounds 2, 4 and 5 require NMR and HRMS spectra since they are new compounds (2 is described in several patents). In the attached file entitled "Manuscript ID, molecules-1360107, Peer Review, Corrections for the main text" the most significant corrections for the text are depicted. 

After implementing these corrections and further corrections that the editor will suggest, in my opinion, the manuscript will be of much higher standard to merit publication in "Molecules". 

Author Response

(The authors gave the same response as above.)

Round 2

Reviewer 2 Report

The authors have explained in good detail the aspects of the in vivo assays and presented evidence of the bioethics review committee. I consider it has improved and my recommendation is to be accepted.